# An approach to linking education, social care and electronic health records for children and young people in South London: a linkage study of child and adolescent mental health service data

Johnny M Downs,[1,2] Tamsin Ford,[3] Robert Stewart,[1,2] Sophie Epstein,[1,2] Hitesh Shetty,[2] Ryan Little,[3] Amelia Jewell,[2] Matthew Broadbent,[2] Jessica Deighton,[4] Tarek Mostafa,[5] Ruth Gilbert,[6] Matthew Hotopf,[1,2] Richard Hayes[1,2]

MH and RH contributed equally.

For numbered affiliations see end of article.

**Correspondence to**
Dr Johnny M Downs;
johnny.downs@kcl.ac.uk

## ABSTRACT

**Objectives** Creation of linked mental health, social and education records for research to support evidence-based practice for regional mental health services.

**Setting** The Clinical Record Interactive Search (CRIS) system was used to extract personal identifiers who accessed psychiatric services between September 2007 and August 2013.

**Participants** A clinical cohort of 35 509 children and young people (aged 4–17 years).

**Design** Multiple government and ethical committees approved the link of clinical mental health service data to Department for Education (DfE) data on education and social care services. Under robust governance protocols, fuzzy and deterministic approaches were used by the DfE to match personal identifiers (names, date of birth and postcode) from National Pupil Database (NPD) and CRIS data sources.

**Outcome measures** Risk factors for non-matching to NPD were identified, and the potential impact of non-match biases on International Statistical Classification of Diseases, 10th Revision (ICD-10) classifications of mental disorder, and persistent school absence (<80% attendance) were examined. Probability weighting and adjustment methods were explored as methods to mitigate the impact of non-match biases.

**Results** Governance challenges included developing a research protocol for data linkage, which met the legislative requirements for both National Health Service and DfE. From CRIS, 29 278 (82.5%) were matched to NPD school attendance records. Presenting to services in late adolescence (adjusted OR (aOR) 0.67, 95% CI 0.59 to 0.75) or outside of school census timeframes (aOR 0.15, 95% CI 0.14 to 0.17) reduced likelihood of matching. After adjustments for linkage error, ICD-10 mental disorder remained significantly associated with persistent school absence (aOR 1.13, 95% CI 1.07 to 1.22).

**Conclusions** The work described sets a precedent for education data being used for medical benefit in England. Linkage between health and education records offers a powerful tool for evaluating the impact of mental health

### Strengths and limitations of this study

► This linkage work sets a precedent for education data being used for patient or medical benefit in England.
► It is one of the few studies that examines linkage errors in children and young people, especially where the non-linked group are not subject to consent-related bias.
► It provides an example of how potential non-random loss between routinely collected health and non-health linked data can be adjusted by weighting techniques.
► Given the constraints of the data available sharing between data controllers, we were unable to assess false-positive matching.
► It was not possible to determine who was not eligible for matching due to complete private or home school educational provision.

on school function, but biases due to linkage error may produce misleading results. Collaborative research with data providers is needed to develop linkage methods that minimise potential biases in analyses of linked data.

## INTRODUCTION

Large-scale longitudinal cohort studies and clinical databases are essential tools for understanding the aetiology and outcomes of childhood mental and physical disorders, including rare or late adverse effects of treatments. However, maintaining the methodological quality of these studies is costly. For example, in the early 1990s, the cost of setting up and sustaining the 15 000 families recruited to Avon Longitudinal Study of Parents and Children birth cohort study (ALSPAC) was around £1 million per

year[1]; few existing longitudinal studies are similarly resourced to sustain representation of their target population.[2] Sample attrition during follow-up can introduce significant methodological biases and can undermine the validity of investigations into novel risk outcome effects.[3] Administrative records from health, education and social public services do not suffer from the same attrition biases by capturing all those receiving a service.[4] They are becoming increasingly available for research: initiatives in Wales and Scotland have now created linked datasets derived from these data resources and are using them to help direct local and national public health strategy.[5]

As yet, the potential gains from these 'big data' systems to drive local population-based analyses to improve child public mental health and educational services remain unrealised in England. Linkage of routinely collected data from public services has the potential to improve how local health, education and social care are delivered to children and young people. Certainly, all mental health services, hospital-based child health services, schools and child protection services that serve the same local area could be more efficient if the design, monitoring, targeting and integration of services were based on data.[6] The ethical and legal processes to do this, as well as the technical security requirements, to gain exemption from individual consent for health data are stringent.[7] Even once these challenges have been met, data matching processes can introduce challenges for health service researchers. For example, the introduction of bias by missed matches, particularly if risk factors are both associated with missed matched records and important outcomes, can impact the validity of research findings derived from linked data.[8] This is more likely to occur when linking routinely collected data via deterministic linkage approaches without a shared identification number (such as health and education records).[9] Deterministic linkage describes an approach when a set of predetermined rules are used to classify pairs of records as matched or non-matched. These tend to require an exact or partial agreement on a set of personal identifiers, for example, a successful match on the first name or surname and match on both the date of birth and postcode. Strict deterministic methods are straightforward to use and commonly employed in government departments; however, they can create high levels of missed matches between records.[10] As a consequence, this undermines the confidence that all the relevant records for an individual have been accurately combined across the different data sources.

In this study, we show how an individual National Health Service (NHS) trust, with coverage of a geographically defined catchment of 1.2 million, ~190 000 children and young people (South London and Maudsley NHS Foundation Trust (SLaM)) developed a sustainable approach to link and anonymise individual children and young people's records from healthcare, social and educational services. We show how a linkage environment that conformed to NHS and Department for Education (DfE)

safeguards was used to build a data resource between an NHS child and adolescent mental health service (CAMHS) records via Clinical Record Interactive Search (CRIS) system[11] linked to the DfE's National Pupil Database (NPD).[12]

This study had two aims: the first was to provide a narrative description of the challenges in gaining approval for a research protocol that needed to meet the legislative requirements for section 251 of the NHS Act 2006, via recommendation from NHS Health Research Authority Confidentiality Advisory Groups,[7] The Education (Individual Pupil Information) (Prescribed Persons) (England) Regulations 2009[13] and subsequent amendments,[14] and satisfy General Data Protection Regulations (GDPR).[15] A second aim was to identify the sociodemographic and clinical factors risk factors, within an NHS CAMHS cohort, that were associated with non-matching with DfE educational records. As an applied example, we used the linked data resource to examine how non-matching may have impacted potential associations between child health factors and school absence (ie, a key education outcome) and how statistical approaches could reduce the effects of this bias.

## METHODS
### The data resources
#### NHS CAMHS data
SLaM provides comprehensive CAMHS to a geographic catchment of approximately 190 000 children and young people resident within four South London boroughs—Croydon, Lambeth, Lewisham and Southwark. SLaM also provides highly specialist services that also accept referrals resident outside the four-borough catchment area. Clinical records have been fully electronic across SLaM services since 2007. The process by which CRIS permits these data to be available for research has been described in detail elsewhere.[6 11 16 17] In brief, CRIS extracts information from the electronic health records generated by CAMH services and, by removing personal identifiers, makes pseudoanonymised data extracts available for analysis by SLaM approved researchers.

CRIS was used to provide an extract of children and young people who were referred to SLaM CAMHS services between 1 September 2007 and August 2013. SLaM has dedicated multidisciplinary services, which assess and treat school age children and young people under International Statistical Classification of Diseases, 10th Revision (ICD-10) multiaxial classification system.[18] The tables and figures within the online supplementary material describe the clinical sample by age and gender first accepted into SLaM CAMHS over a 5-year period (online supplementary table 1 for ICD-10 rates in the clinical sample). As online supplementary figures 1 and 2 shows, the majority of children and young people are first seen in CAMHS services in midchildhood and will often receive short discreet periods of care. However, some will receive prolonged CAMH services throughout child and adolescence.

## Department of Education National Pupil Database

The NPD is a pupil-level longitudinal database that matches pupil and school characteristic data to pupil-level attainment.[12] The key datasets within the NPD are the pupil census and pupil attainment datasets, which hold data for all assessments that pupils complete during primary and secondary school state education. The NPD pupil census provides a snapshot of pupils attending state-maintained schools in England ~91% of pupils resident with the SLaM catchment,[19] which is submitted annually on a specific day in January, by a school for all pupils in that school. Pupils held within the NPD are typically aged between 3 years and 19 years but some from special schools may be up to age 24 years.

### The technical resources

To link CRIS data with other external clinical and non-clinical sources, SLaM developed a research governance model for linking data that satisfy NHS requirements as described in Department of Health Information Governance Review, or 'Caldicott 2' report.[20] In accordance with these guidelines, SLaM set up the Confidential Data Linkage Service (SLaM CDLS)[11] as a Trusted Third Party or Safe Haven to ensure that confidential patient information can be linked in a way that guarantees the legal and ethical rights of patients and caregivers. A similar provision was available in DfE Data Services Provision, which had a linkage service, governed under HMG Security Policy Framework v10 2013,[21] with experience of regularly undertaking external linkages with large scale research cohorts including the Millennium Cohort Study[22] and ALSPAC.[1]

### Linkage

#### Preparing the CRIS CAMHS identifiers for matching

We selected a cohort of young people aged between 4 years and 18 years who were referred to SLaM mental healthcare between 1 September 2007 and 31 December 2013. As described previously, in the UK, unique identifiers, such as national health identifiers, are not shared between health and education databases, so records require matching on personal identifiers common to both data resources (ie, names, dates of birth and residence postcode).

Personal identifiers were standardised using the following definitions:
1. **Dob**: format (dd-mm-yyyy).
2. **forename_1**: the first word present in the forename field registered for the individual record (ie, all text left of the first white space character in the free-text field).
3. **forename_2**: the second word present, if >1 forename present (ie, second of 2+ names separated by one space or punctuation except '-') (ie, right of white space).
4. **surname_1**: the first word present in the surname field registered for the individual record (ie, all text left of the first white space character).
5. **surname_2**: the second word present, if >1 surname present (ie, second word of 2+ names if separated by one space or punctuation except '-').
6. **surname_3**: the whole string in the surname field.

Within the longitudinal health record, there were often several different addresses held for each individual. Similarly, there were multiple addresses held for most pupils in the education database. Pupil address data are routinely updated on the 16 January every year. So, we developed a hierarchical system to extract the postcode from the health record most likely to match with education database. Figure 1 shows how this postcode hierarchy might be applied to one individual child, where the blue blocks represent episodes of care provided by CAMHS, and the green time line represents the period of time in school. Taking these considerations into account, we produced a hierarchy of postcodes with 1–5 levels for each individual seen in CAMHS using logic rules (see figure 1 legend).

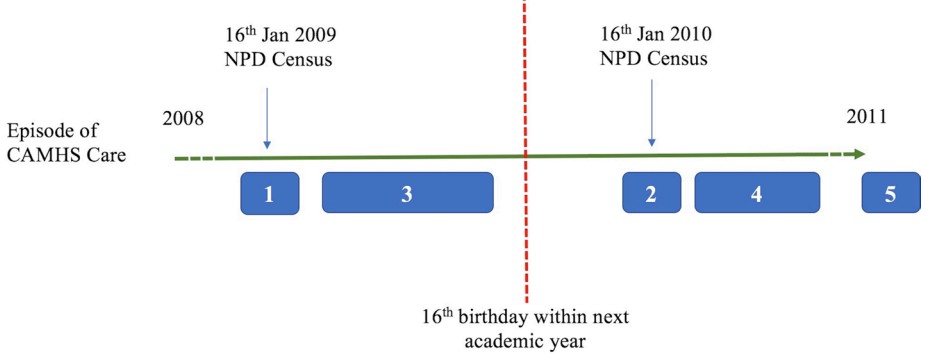

**Legend:** Postcode address hierarchy 1-5 provided for matching by SLaM to NPD

1. SLaM recorded address most likely to coincide within school census before age 16 years
2. SLaM recorded address most likely to coincide within school census before age 18 years
3. SLaM recorded address held for the greatest duration before age 16 years
4. SLaM recorded address held for the greatest duration before age 18 years
5. Any available postcode recorded by SLaM where 1-4 not available

*NB: numbers within the blue blocks represent residential address postcodes according to the legend presented above

**Figure 1** Creating a hierarchy of matching postcodes to improve the link between CRIS CAMHS data to DfE National Pupil Database (NPD). CAMHS, child and adolescent mental health service; CRIS, Clinical Record Interactive Search; DfE, Department for Education; SLaM, South London and Maudsley NHS Foundation Trust.

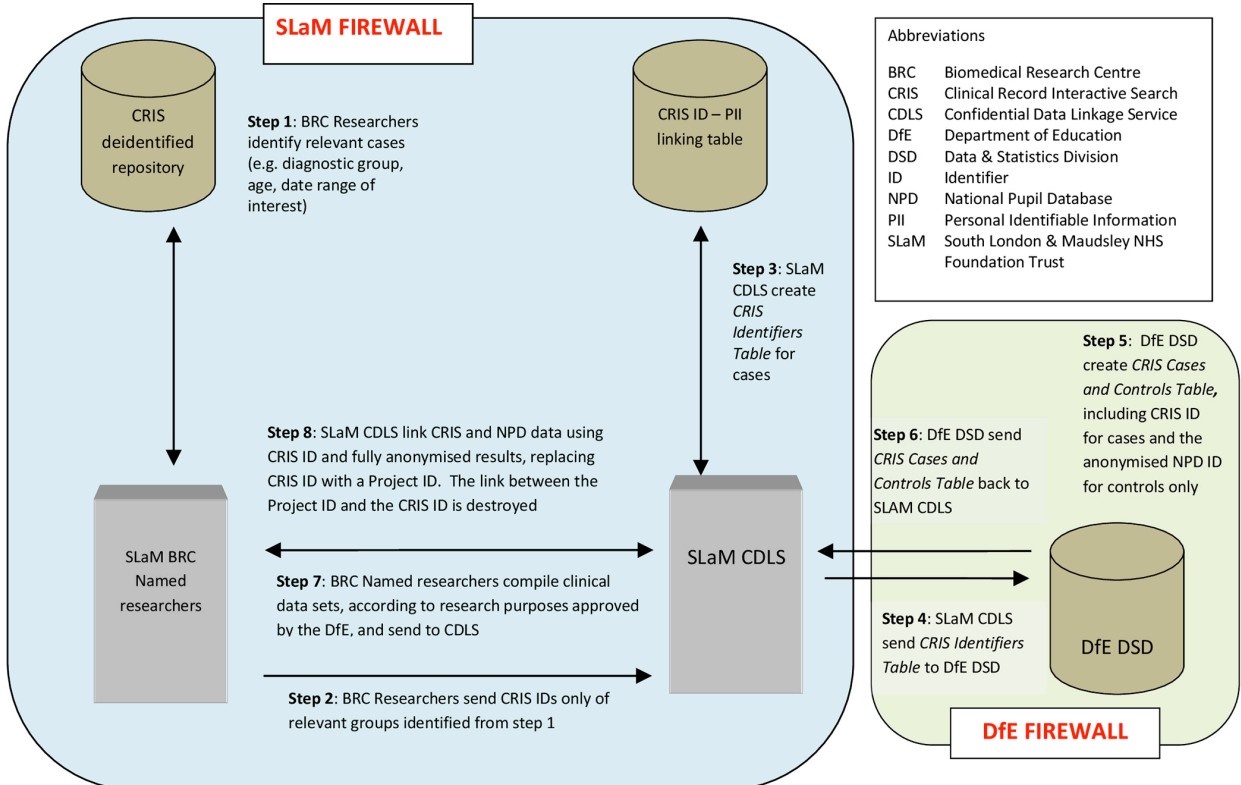

**Figure 2** Data flow process linking CRIS CAMHS Data to the National Pupil Database.

A SQL-based query was used to extract the identifier data according to these rules. This produced a sample of 36 760 individuals with distinct individual records. Postextraction, we then ran data cleaning and logic checks that included removal of all those with numbers in name string fields (four cases removed), all those with only one letter in their first or surname (one case removed) and all those with incomplete/atypical English postcodes (214 records hand searched, 77 valid English postcodes were cleaned and retained). We excluded all children whose first referral date was less than 4 years (1095 days) after their date of birth, unless they had confirmed follow-up contact details recorded within the window (ie, 2007–2013) at least 1 year later than the earliest referral date. This was because clinicians can erroneously record the date of referral or time seen at initial appointment in the date of birth field. This mainly occurs in individuals with only single episodes of contact with services. To fit in with the academic calendar and UK school age, children were then selected if they had their fourth birthday prior to the 1 September 2012. This provided a complete sample of 35 509 ready for matching with the NPD.

All the data prepared for matching had personal identifier fields populated with the exception of the secondary surnames and forenames (ie, there were no missing values). Dates of birth ranged from 6 January 1989 to 31 August 2008, which meant that all of these pupils could potentially be found in either current or historic NPD census data. Personal identifiers were standardised to maintain a consistent format with NPD identifiers:

SLaM identifiers were prepared to fit with DfE first name, surname and date of birth formats, which included standardising string length, capitalisations, use of spaces and hyphens.

Only identifiers (names, postcode and date of birth), accompanied by their unique CRIS ID pseudonym, were then sent via secure file transfer to the DfE Data and Statistics Department.

As represented in figure 2 (and described in four stages below), the DfE matched these against NPD personal identifiers (approximately 15 million records), generating a pupil-specific, non-identifiable NPD ID variable across the whole data set and adding the CRIS ID to this table for cases only, stripping the resultant table of all identifiers other than the anonymised NPD ID and the pseudonymised CRIS ID and transferring the data set back to SLaM CDLS using a secure file transfer.

The supplied data items by the SLaM CDLS were matched to the NPD data by DfE informaticians in the stages described below. Initial matching or stage 1 was based on exact matches for the supplied data items. For SLaM cases who did not meet stage 1 matching criteria, stage 2, 'fuzzy' matching processes were conducted, and so on, down to stage 4.

► Stage 1: full match on any combination of CAMHS names (all supplied values including alias), dates of birth and postcode (all supplied) were conducted against the most recent address held, and then the working back, all years/terms of the school census data, pupil referral data, alternative provision data,

early years census data. School census data contained preferred and former surnames, which were also searched. Forenames were checked against forename/middle name combinations.

► Stage 2: full match on date of birth, postcode and fuzzy matching on names. To ensure confidence in these matches, results were checked manually. Fuzzy matching was conducted on first two characters of names.

► Stage 3: full match on names and dates of birth, postcode inward code (the first 2–4 characters) plus first character of the outward code (the latter characters after the space). To ensure confidence in these matches, results were checked manually.

► Stage 4: full match on names and postcode with manual check of dates of birth, looking for 'near' dates of birth – where the record may be possibly 1 year out, 1 month out, 1 day out and transposed month/day.

### Analysis of linkage bias

Overall linkage rate was calculated as the percentage of CAMHS individuals linked to any NPD school record on any of the stages 1–4. Potential sources of linkage biases were estimated by comparing linked and unlinked data. For the CAMHS sample described in table 1, we categorised an individual match to NPD school absence data (a subset of the NPD school record) as a binary outcome: match=1, non-match=0. The ICD-10 multiaxial classification system[18] was used to categorise the presence of any recorded mental health diagnosis (ie, diagnoses status prior to 18th birthday) available between 2007 and 2013.

Using multivariable logistic regression, we explored the associations between a number of risk variables including demographic (eg, gender, ethnicity and neighbourhood deprivation), clinical (age at first presentation to CAMHS and diagnosis of any ICD-10 disorder) and administrative factors (eg, postcode hierarchy; see figure 1) with linkage to the school attendance database as the binary outcome. We used this logistic regression to generate a probability estimate of matching as a function of the risk variables.

### Patient and public involvement

In terms of gathering evidence for support of the public benefit to use patient identifiable data via CRIS to link to the NPD without patient or caregiver consent, we consulted several clinical, patient and caregiver groups. We invited comments on privacy notices, gave presentations and collected minutes from the SLaM child and adolescent psychiatry executive group, the Service User Research Enterprise group, National Young Persons Advisory Group, the service user-led CRIS Oversight Committee and SLaM-involved parents, through the Biomedical Research Centre (BRC) patient engagement programme.[23] Because of the focus of one of the projects using the linked data was an investigation into the educational outcomes of children and young people with autism spectrum disorders, we also invited comments

on the proposal from the National Autistic Society. A lay summary of the purpose of the data linkage was written in collaboration with the Maudsley National Institute for Health Research (NIHR) service user data linkage advisory group (eg, https://www.maudsleybrc.nihr.ac.uk/facilities/clinical-record-interactive-search-cris/cris-data-linkages/), and a short video was made to raise awareness of the study and future research plans.

### Analysis of linkage error using school attendance outcomes

It is challenging to assess the impact of linkage error for a particular outcome, when there is not an expected one-to-one relationship between one variable and another. For example, when linking patient records to a death registry to determine a patient's survival status, it is difficult to know which matches have been missed; the death registry will only contain patients who have died, and so a non-match could be due to patient being alive or being a missed match.[24] Applying this to school data, there was a need to select a clinically relevant school performance outcome that should be available for all pupils. School absence was chosen as the outcome to assess linkage error because school attendance is a clinically relevant and systematically recorded for all pupils accessing state school.

For each matched CAMHS-NPD pupil, a binary outcome marker of poor attendance was created for the latest academic year they attended school available between 2007/200808 and 2012/2013. Pupils were categorised as persistent absentees if they had recorded less than 80% school attendance for the total number of possible school sessions available since their enrolment for that academic year (one session is equal to half a school day).

Using the probability of matching estimate from the linkage bias analysis, we created a weight that was inversely proportional to the probability of being linked to NPD school attendance data, which was assigned to each individual with linked CAMHS school absence data. This followed standard methodology for managing non-response bias in conventional cohort and survey designs.[25] Multivariable logistic regression was used to examine predictor variables and association with persistent school absence, initially without weights, and then with inverse probability weights. To examine another approach to adjust for potential selection bias from non-linkage,[26] we examined whether the main effects of interest also persisted after the probability of matching estimate was entered as a covariate in the multivariable logistic regression model.

## RESULTS

### Section 1: achieving the ethical, governance and legal approvals

The proposal to link the NPD and CRIS CAMHS data underwent a robust and lengthy ethical, legal, governance and technical review, conducted by a number of local and national committees within NHS and DfE.

**Table 1** Sociodemographic characteristics of the child and adolescent mental health sample linked and non-linked to the National Pupil Database absence data

| | Linked pairs (n=29 278) (%) | Non-linked residuals (n=6231), n (%) | OR (95% CI) for positive linkage | aOR (95% CI)† |
|---|---|---|---|---|
| Male | 16 430 (56.1) | 3296 (52.9) | *Reference* | *Reference* |
| Female | 12 848 (43.9) | 2935 (47.1) | 0.88 (0.83 to 0.93)** | 1.04 (0.97 to 1.11) |
| Age at first referral to mental health services | | | | |
| Infant (<7 years) | 3657 (12.5) | 535 (8.7) | *Reference* | *Reference* |
| Primary (7–11 years) | 10 980 (37.5) | 1284 (20.3) | 1.25 (1.12 to 1.39)** | 1.23 (1.10 to 1.38)** |
| Secondary (12–15 years) | 7048 (24.1) | 1140 (18.4) | 0.90 (0.81 to 1.01) | 0.98 (0.88 to 1.10) |
| College (16–18) | 7570 (25.9) | 3228 (52.2) | 0.34 (0.31 to 0.38)** | 0.67 (0.59 to 0.75)** |
| Ethnicity | | | | |
| White/white British | 13 838 (47.3) | 2786 (44.7) | *Reference* | *Reference* |
| Asian/Asian British | 984 (3.4) | 312 (5.0) | 0.63 (0.56 to 0.76)** | 0.65 (0.56 to 0.75)** |
| Black British/African | 5667 (19.4) | 1181 (19.0) | 0.96 (0.89 to 1.04) | 0.82 (0.76 to 0.89)** |
| Black British/Afro-Caribbean | 1474 (5.0) | 232 (3.7) | 1.28 (1.11 to 1.48)** | 0.98 (0.84 to 1.14) |
| Mixed/multiple ethnicity | 2184 (7.5) | 315 (5.1) | 1.40 (1.23 to 1.58)** | 1.12 (0.99 to 1.28) |
| Other ethnic group | 1109 (3.8) | 419 (6.7) | 0.53 (0.47 to 0.60)** | 0.55 (0.48 to 0.63)** |
| Not stated | 4022 (13.7) | 986 (15.8) | 0.82 (0.76 to 0.89)** | 0.93 (0.85 to 1.02) |
| Resident within Local catchment area | 22 481 (76.8) | 4192 (67.2) | 1.61 (1.52 to 1.71)** | 1.04 (0.97 to 1.12) |
| National quartiles of neighbourhood deprivation | | | | |
| First (most deprived) | 14 398 (49.2) | 2822 (45.3) | *Reference* | *Reference* |
| Second | 9796 (33.5) | 2179 (34.9) | 0.88 (0.83 to 0.94)** | 0.90 (0.83 to 0.96)** |
| Third | 2956 (10.1) | 762 (12.2) | 0.76 (0.69 to 0.83)** | 0.81 (0.74 to 0.89)** |
| Fourth (least deprived) | 2126 (7.3) | 468 (7.5) | 0.89 (0.79 to 0.99)* | 1.03 (0.92 to 1.15) |
| Address data available‡ | | | | |
| Postcode 1 | 17 587 (60.1) | 1987 (31.9) | *Reference* | *Reference* |
| Postcode 2 | 2956 (10.1) | 990 (15.9) | 0.34 (0.31 to 0.37)** | 0.50 (0.45 to 0.56)** |
| Postcode 3 | 5776 (19.7) | 1187 (19.1) | 0.55 (0.51 to 0.59)** | 0.63 (0.58 to 0.68)** |
| Postcode 4 | 1933 (6.6) | 1010 (16.2) | 0.22 (0.20 to 0.23)** | 0.35 (0.31 to 0.39)** |
| Postcode 5 | 1026 (3.5) | 1057 (17.0) | 0.11 (0.09 to 0.12)** | 0.15 (0.14 to 0.17)** |
| Any ICD-10 disorder | 17 749 (60.6) | 3290 (52.8) | 1.38 (1.30 to 1.45)** | 1.11 (1.04 to 1.18)** |

*P<0.05, **p<0.01.
†Adjusted for all other covariates listed in the table.
‡Postcode. For a large proportion of cases, there are several addresses available for each case. Therefore, postcodes were extracted according to a hierarchy (postcode 1 being the highest), which we believed to be most likely to have been the place of residence on the day of the 16 January 20XX (variable date) census (see figure 1 legend).
aOR, adjusted OR; ICD-10, International Statistical Classification of Diseases, 10th Revision.

Figure 3 provides the timeline and milestones achieved to reach the completion of the linked DfE-SLaM CAMHS dataset. We provide in-depth description of the process as an online supplementary reports to this paper. In brief, gaining the permissions to link the NPD and CRIS CAMHS data was complex, as there was no precedent in England for such a linkage between routinely collected mental health and school data, and there had been no successful completion of linked NHS and non-NHS non-health data without individual consent.[27] After a round of discussions between DfE and SLaM, we described a process to link the data, with the main research purpose focused on estimating the effects of clinically recognised, mental health disorder and treatment on educational outcomes. Research governance approval was granted by the SLaM Caldicott Guardian Committee and the DfE's Data Management Advisory Panel in principle, but the linkage process was contingent

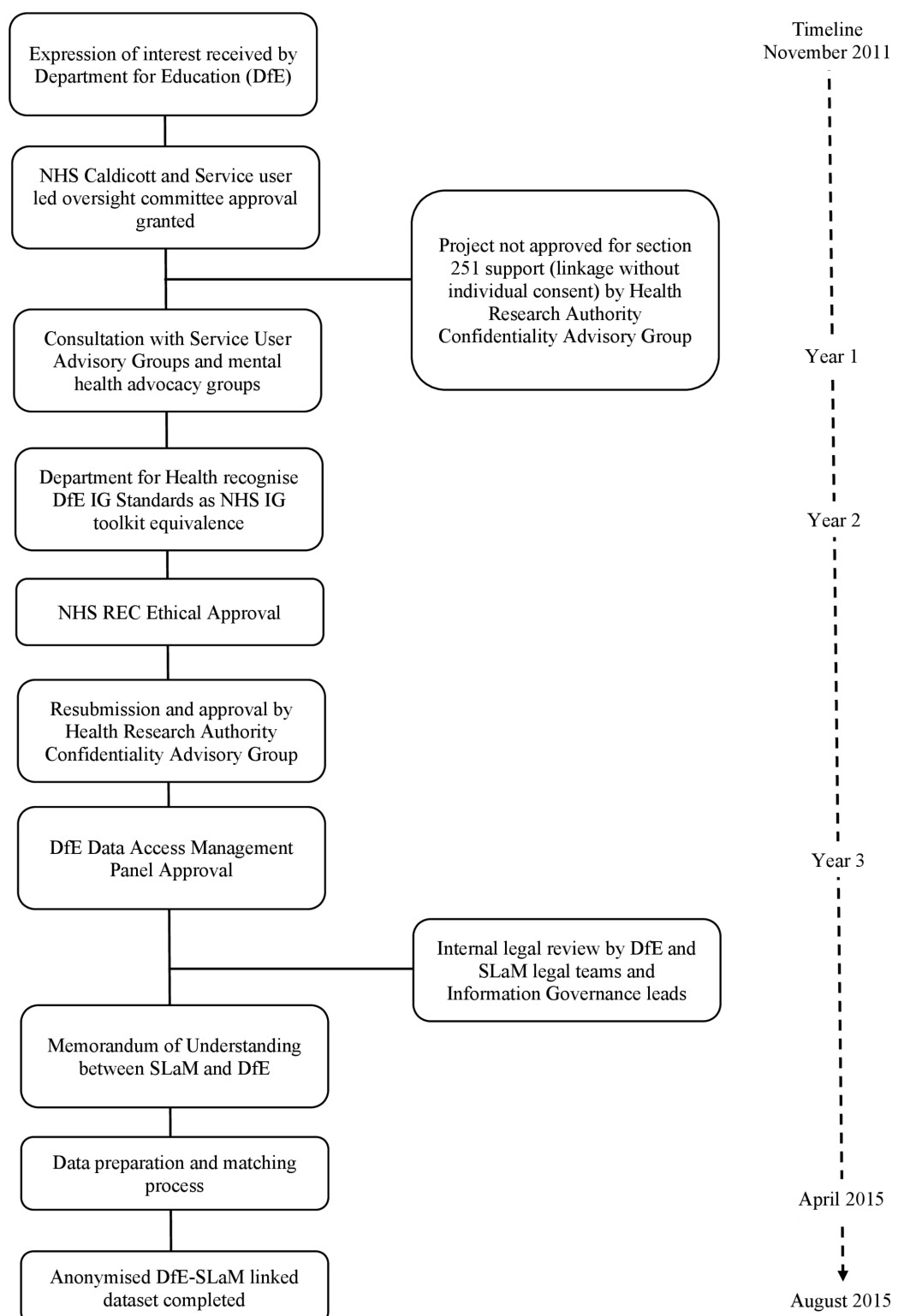

**Figure 3** A timeline of the ethical, legal and technical milestones for reaching a data linkage between DfE and SLaM. DfE, Department for Education; IG, information governance; NHS, National Health Service; REC, Research Ethics Committee; SLaM, South London and Maudsley NHS Foundation Trust.

on Health Research Authority Confidentiality Advisory Group (HRA CAG) approval.[7]

The HRA CAG rejected the first application, as the research activity proposed did not demonstrate sufficient medical purpose and public benefit to meet the s251

requirements (please see online supplementary report for further details). Research conducting a longitudinal analyses of health exposures on education outcomes was not sufficient to meet criteria for conducting research for medical purpose. The HRA CAG also queried whether

linkages could not be better carried out using NHS Digital's Trusted Data Linkage Service. The CAG advised that this route would negate the requirement for SLaM to disclose confidential patient information to the DfE and minimise the disclosure of patient information. A final major issue related to the governance arrangements in place around the processing of patient data by the DfE. We had not provided sufficient information around retention periods, access arrangements and the extent of identifiable data requested. To prepare for resubmission, we revised the scientific proposal to be more focused on understanding the bidirectional associations between educational performance and mental health disorders. To gather more evidence the public benefit case of scientific proposal, we involved our local NIHR BRC patient and clinician engagement programme, relevant charitable and education sector bodies.

To address the second issue, we acknowledged that an additional potential benefit to using NHS Digital was that patient identifiers would be retained within an NHS environment, but we were able to confirm that both the SLaM CDLS and DfE were in line with government standards and meet equivalent to information governance (IG) expectations for NHS care system organisations.[28]We also demonstrated, by reviewing the alternative data flows, that using NHS Digital as the trusted third party in this linkage would prove a more complex, and less secure linkage method (please see online supplementary reports). Briefly, both DfE and SLaM data controllers expressed concern that the additional step of involving NHS Digital would significantly increase the potential risk of harm if a breach of data security occurred, especially given the scale and sensitivity of the educational data and the very large number of individuals involved (over 15 million children).

### Section 2: linkage rates, bias and the impact on education outcome analyses

The overall matching process against any NPD attendance records provide 29 278 CAMHS-NPD linked records representing a linkage rate of 82.5%. The proportions linked according to DfE matching stages described above: stage 1%–60.2%, stage 2%–4.2%, stage 3%–1.2% and stage 4%–16.9%.

Table 1 identifies the SLaM CAMHS sociodemographic, clinical and administrative record risk factors for linkage to the NPD data. An OR greater than 1 denotes greater chance of successful linkage compared with the reference. In the adjusted model, we found significant differences in most sociodemographic, clinical and administrative factors. Compared with school age children aged under 7 years, children first referred to CAMHS in late adolescence were significantly less likely to be matched to the NPD (OR 0.67, 95% CI 0.59 to 0.75, p<0.01), while children aged 7 to 11 years, were more likely to be successfully matched (OR 1.23, 95% CI 1.10 to 1.38, p<0.01). Relative to children of white ethnicity, we found other ethnic groups including Asian, black African and mixed groups were less likely to be matched. There were no significant differences

in successful linkage between children and young people in the lowest and highest quartiles of deprivation, but there was significantly reduced linkage success for those living in neighbourhoods in the second and third quartiles. Analyses of the administrative characteristics show that the postcodes (which were extracted from clinical episodes of care and did not overlap with January census data (ie, postcodes 2, 4 and 5; see figure 1) were less likely to link even after adjustment for other potential explanatory variables (see table 1).

Table 2 provides the sociodemographic, clinical and administrative record characteristics for children and young people seen SLaM CAMHS and the associated risk for persistent absence. The adjusted analyses show that presence of an ICD-10 mental health disorder (aOR 1.13, 95% CI 1.07 to 1.22, p<0.01), age at first referral to CAMHS and mixed ethnic group (relative to white ethnic groups) were associated with an increased risk of persistent school absence, while Asian, black African and Black Caribbean ethnicity, increased neighbourhood affluence was associated with a decreased risk of persistent absence. These effects persisted after both statistical techniques (1) using inverse probability weighting and (2) adjustment for matching probability were applied to reduce matching bias in the adjusted analyses.

### DISCUSSION

We provide the first example of how data linkage projects can be completed using routinely collected NHS and DfE educational data. This use case demonstrates how the legal basis for the 'public benefit' (ie, without individual level consent) can be made to satisfy GDPR.[15] The regulatory and technical issues for data sharing between health and non-health services are challenging in England but surmountable. Using deterministic matching techniques provided by the DfE, a large-scale dataset was built between NHS child and mental health data and national school administrative data, providing a linkage for 29 278 patients (82.5% of the NHS cohort) to their educational records. There were significant differences in the sociodemographic and clinical characteristics between matched and non-matched NHS samples. Using these data, we found any child or young person with an ICD-10 mental disorder had approximately 10% greater likelihood of having persistent school absence, when compared with those clinically referred and not meeting threshold for diagnosis. Effects did not change significantly after matching probability adjustment, which suggests these effects on were not driven by selection bias from matching errors.

### Analysis of the linkage biases

Overall, we found 17.5% of the clinical population were not successfully matched to NPD absence data. While enrolment at a non-state maintained school or independent school may explain a proportion,[6 19] a significant minority were likely to match due to administrative factors, which may include missingness or inconsistencies of the matching identifiers, as demonstrated by the effect of postcode

**Table 2** Sociodemographic and ORs for persistent (>80%) school absence in 29 278 children and young people referred to mental health services

| | No persistence absence (n=23 241), (%) | Persistent school absence (n=5635), n (%) | OR (95% CI) | aOR† (95% CI) | Weighted aOR‡ | Match probability adjusted aOR§ |
|---|---|---|---|---|---|---|
| Any ICD-10 disorder | 14 004 (60.2) | 3594 (63.7) | 1.16 (1.09 to 1.23)** | 1.13 (1.07 to 1.22)** | 1.13 (1.07 to 1.22)** | 1.10 (1.03 to 1.19)** |
| Age at first referral to mental health services | | | | | | |
| <7 years) | 3031 (13.0) | 298 (5.3) | Reference | Reference | Reference | Reference |
| 7–11 years | 9405 (40.5) | 1540 (27.3) | 1.67 (1.46 to 1.90)** | 1.67 (1.46 to 1.90)** | 1.67 (1.47 to 1.91)** | 1.60 (1.49 to 1.84)** |
| 12–15 years | 5205 (22.4) | 1830 (32.5) | 3.58 (3.14 to 4.07)** | 3.65 (3.20 to 4.18)** | 3.71 (3.24 to 4.23)** | 3.66 (3.21 to 4.18)** |
| 16–18 years | 5600 (24.1) | 1967 (34.9) | 3.57 (3.13 to 4.06)** | 4.20 (3.63 to 4.86)** | 4.15 (3.57 to 4.81)** | 4.70 (3.82 to 5.78)** |
| Female | 10 023 (43.1) | 2695 (47.8) | 1.20 (1.14 to 1.28)** | 0.97 (0.91 to 1.03) | 0.97 (0.92 to 1.04) | 0.96 (0.91 to 1.03) |
| Ethnicity | | | | | | |
| White/ white British | 10 651 (45.8) | 3011 (53.4) | Reference | Reference | Reference | Reference |
| Asian/ Asian British | 815 (3.5) | 159 (2.8) | 0.69 (0.58 to 0.82)** | 0.68 (0.57 to 0.81)** | 0.69 (0.58 to 0.83)** | 0.76 (0.60 to 0.96)* |
| Black British/ African | 4737 (20.4) | 849 (15.1) | 0.63 (0.58 to 0.69)** | 0.68 (0.62 to 0.74)** | 0.69 (0.63 to 0.75)** | 0.71 (0.64 to 0.79)** |
| Black British/Afro-Caribbean | 1213 (5.2) | 248 (4.4) | 0.72 (0.63 to 0.83)** | 0.81 (0.70 to 0.94)** | 0.81 (0.70 to 0.94)** | 0.82 (0.70 to 0.94)** |
| Mixed/multiple ethnic | 1653 (7.1) | 483 (8.6) | 1.03 (0.93 to 1.15) | 1.14 (1.02 to 1.28)* | 1.15 (1.03 to 1.29)* | 1.11 (0.99 rto1.26) |
| Other ethnic group | 905 (3.9) | 195 (3.5) | 0.76 (0.64 to 0.89)** | 0.78 (0.66 to 0.92)** | 0.80 (0.67 to 0.96)** | 0.92 (0.69 to 1.22) |
| Not stated | 3286 (14.1) | 694 (17.4) | 0.74 (0.68 to 0.82)** | 0.78 (0.71 to 0.86)** | 0.79 (0.72 to 0.87)** | 0.79 (0.72 to 0.87)** |
| Resident within Local catchment area | 18 100 (77.8) | 4064 (72.1) | 0.74 (0.69 to 0.76)** | 0.88 (0.82 to 0.95)** | 0.89 (0.83 to 0.96)** | 0.87 (0.80 to 0.94)** |
| National quartiles of neighbourhood deprivation | | | | | | |
| First (most deprived) | 11 326 (79.7) | 2884 (51.1) | Reference | Reference | Reference | Reference |
| Second | 7891 (33.9) | 1785 (31.7) | 0.89 (0.83 to 0.94)** | 0.83 (0.76 to 0.89)** | 0.82 (0.77 to 0.88)** | 0.85 (0.79 to 0.92)** |
| Third | 2349 (10.1) | 557 (9.9) | 0.93 (0.84 to 1.03) | 0.74 (0.69 to 0.83)** | 0.74 (0.66 to 0.83)** | 0.78 (0.69 to 0.89)** |
| Fourth (least deprived) | 1692 (7.3) | 413 (7.3) | 0.96 (0.85 to 1.07) | 0.70 (0.62 to 0.80)** | 0.70 (0.62 to 0.80)** | 0.69 (0.62 to 0.78)** |
| Address data available¶ | | | | | | |
| Postcode 1 | 14 119 (60.7) | 3170 (56.2) | Reference | Reference | Reference | Reference |
| Postcode 2 | 2287 (9.8) | 669 (11.9) | 1.30 (1.18 to 1.43)** | 0.71 (0.63 to 0.78)** | 0.71 (0.64 to 0.81)** | 0.85 (0.65 to 1.11) |
| Postcode 3 | 4618 (19.9) | 1077 (19.1) | 1.03 (0.96 to 1.12)** | 0.92 (0.84 to 0.99)* | 0.92 (0.85 to 1.00) | 1.01 (0.87 to 1.19) |
| Postcode 4 | 1448 (6.2) | 485 (8.6) | 1.49 (1.33 to 1.67)** | 0.81 (0.71 to 0.93)** | 0.82 (0.72 to 0.95)** | 1.14 (0.71 to 1.81) |
| Postcode 5 | 788 (3.4) | 238 (4.2) | 1.34 (1.16 to 1.56)** | 0.93 (0.79 to 1.10) | 0.93 (0.78 to 1.09) | 1.85 (0.74 to 4.66) |

*P<0.05, **p<0.01.
†Adjusted for all other covariates listed in the table.
‡Adjusted model with inverse probability weighting for matching included.
§Adjusted model with addition of matching probability estimates entered as a covariate.
¶See figure 1 legend.
aOR, adjusted OR; ICD-10, International Statistical Classification of Diseases, 10th Revision.

variation in the analysis, or errors secondary to the matching process. There have been very few studies conducted that examine linkage errors in children and young people, especially where the non-linked group are not subject to consent related bias. Previous research suggests that ethnic minorities are more likely to have administrative records with misspelt names, inaccurately recorded dates of births and higher levels of residential instability, which may be applicable to this sample.[9 29] These findings provide further argument for greater collaborative research with data providers to develop linkage methods that minimise potential biases in analyses of linked data.[10] Deterministic processes that offers little flexibility in matching misspelt names may be a reason why ethnic variation may contribute to missed matches.[9] We found certain age groups, particularly those aged 7–11 years, were associated with a greater

likelihood of linkage. This may be due to the greater availability of accurate personal identifiers in the records of this group, as their potential exposure to CAMHS services while at school will be longer than other age groups. Similarly, having a ICD-10 mental disorder, which also had an increased likelihood of linking with the school data, may be related to identifier accuracy, as higher levels of psychopathology are associated with greater clinical contact, and potentially higher clerical accuracy in recording personal identifiers. It is also more probable that those with higher levels of psychopathology will have longer durations of care that overlap with the school census date.

We found a U-shaped distribution in neighbourhood deprivation and likelihood of linkage. Compared with areas with the highest deprivation, areas within the second and third quartiles showed significantly reduced likelihood of linkage, but the most affluent areas showed minimal difference. This could relate to families from affluent areas being able to comply with the administrative process, and/or correct administrative errors, and families from the highest deprived areas having greater need and hence higher clinical contact with services. Both these factors may improve clerical accuracy and concordance with school data. Families from second and third quartiles may have less of both these characteristics and hence reduce their likelihood of linkage. The current data available in this study does not permit this hypothesis to be tested, but findings suggest that a more detailed extraction examining frequency of clinical contact with services and data linkage outcome is an area for future work.

In our sample, linkage biases appear to have little effect on the association between mental disorder and attendance. However, without information from source data, potential linkage error could be introduced without researchers being aware whether there was need for it to be accounted for in subsequent analyses. Our study highlights the importance of governance arrangements between linkers and analysts to identify which groups are disproportionately affected by linkage error. In our case, by permitting approved NHS researches to examine the identifier fields of matched and unmatched SLaM samples, this governance has enabled some flexibility with the 'data separation principle': a common practice in data linkage research, where identifiers (eg, names or date of birth) are kept separate from attributes (in this case health or education data), to protect privacy and avoid disclosure during the linkage process.[30] While the separation principle might reduce the risk of identification, it does not permit researchers to evaluate the potential risk of linkage bias on future analyses.

### Implementation challenges to the data linkage between health and education data

We believe the tasks and challenges to use personal health and education data for data linkage and research can be best described as 'establishing the social license'.[31] This activity included articulating a clear purpose for the linkage, recognised as beneficial by the public or those potentially involved as data subjects and that the potential risks to individuals or public institutions were tolerable in relation to these benefits. Without the evidence of the proposal being scrutinised and ultimately accepted by those potentially involved as data subjects, and the public institutions/services who act as controllers of the data, it would have been difficult to sustain a case for public benefit; in fact, this was one of the reasons why the first application was not approved by the HRA CAG. To prove we had *social licence* to conduct the linkage work, we needed to gather evidence from a number of sources including service users, clinicians, academics, advocacy groups and governance leads, all who may have had a stake in the process and outcomes of the data linkage project.

The second aspect of establishing the social licence to conduct the linkage work, related to fulfilling the professional mandate for properly conducting the linkage process and related research activity. This involved making sure the proposal complied with the known legal, technical and ethical frameworks that governed health data use and any additional safeguards deemed important by the data controllers and custodians. The technical aspects were not just confined to data security but also involved preparing the data to ensure the most accurate match to reduce error and redundancy in later analysis. Fulfilling the mandate also involved the creation of formal contract between the parties involved in controlling, sharing, processing and using the data. This mandate committed us to conduct appropriate analysis and dissemination of the linkage related research, so that we could sustain the social licence for future research activity. This may be especially pertinent in England as linkage-driven research of routinely collected public service activity is in its infancy and benefits are yet to be comprehensively established.

Given the time and resources spent to set up this linked data resource, and the potential it holds, it is important that these resources are maintained and remain accessible for reuse in the future. Without developing specific data sharing agreements between the parties, it can be difficult to establish a collaborative relationship with good governance structures between the controllers, linkers and analysts. Without these structures, there may be a tendency for data controllers to agree to link data only via a 'create and destroy' approach. We believe this maybe unethical in terms of waste and scientifically unsound as prior analyses cannot be re-examined. It also re-exposures data subjects to the potential risks of sharing personal identifiable information again across different agencies should the linkage need to be repeated in the future.

### Strengths and limitations of the matching methods and matching evaluation

This study has a number of strengths. First, it presents a novel application to link data across public sector organisations. The description of the legal, ethical and technical challenges and solutions are described to share some of the lessons we have learnt through the process, in the hope that they will be useful for other public organisations. Furthermore, the

study provides an example of how potential non-random loss between routinely collected health and non-health linked data can be adjusted by weighting techniques. Because the source data was available to examine missed linkages, we were able to determine that linkage error did not lead to systematic biases and misleading positive estimates between ICD-10 mental disorder and persistent school absence. The demonstration of matching probability adjustment and inverse probability weighting was intended to illustrate how linkage bias may be reduced, not as a definitive analysis of these data. Given its policy relevance, we reported on a single categorical absence outcome, less than 80% annual school attendance. Whether the same associations hold for other discrete levels of absence (eg, 60% or 90%) certainly warrants examination in future analyses. Examining methods to improve linkage techniques, coupled with newer methods for handling uncertainty in analysis of linked data, should also help improve the generalisability and quality of future population-based linkage studies.[27]

The matching methods in this study have a number of limitations. We were unable to assess false-positive matching nor able to assess risks for the lower confidence matching (DfE stages 2–4, described above) and the potential effects on school outcome analyses. No shared unique identifier exists between NHS and educational services nor were their governance arrangements or sufficient resources in place to manually compile an NPD-SLaM CAMHS-linked gold standard data. Another limitation of the matching methodology is the limited number of address identifiers that could be used. For example, due to governance constraints, we were unable to use first line of the address, which again limited the capacity to potentially check for coding errors in the postcode. Another contributing factor to linkage error was the age of the child. A substantial number of young people were seen in CAMHS aged 16 years and 17 years and would not have data on the NPD if they were no longer attending school. Similarly, we were unable to determine who was not eligible for matching due to complete private or home school educational provision that may be, at greatest, 10% of the sample. These limitations are likely to have led to our finding being an underestimation of the linkage performance.

The matching evaluation also has several limitations. We only reported on a single categorical absence outcome (less than 80% annual school attendance); whether linkage error had similarly limited effects on other discrete levels of absence (eg, 60% or 90%) was not evaluated. ICD-10 codes permitted us to evaluate the effect of reaching threshold for a 'clinical disorder' on absence rates in an efficient and cost-effective manner. However, collapsing ICD-10 categories into one binary variable only provided an 'average' effect across all ICD-10 diagnoses. This may have introduced aggregation bias, which disguised the potential heterogeneity of effects across different the diagnoses. Furthermore, the validity of ICD-10 codes in psychiatric registers can be variable, and although we did not disaggregate ICD-10 cases into specific disorders, it is known that some disorder codes are more likely to be misclassified than others, or at least

more prone to diagnostic revision.[32] Assessing the effect of variation in ICD-10 validity on school outcomes was beyond the scope of this study. However, we have provided solid groundwork for future research to refine the characterisation of clinical phenotypes either via algorithms that offer greater diagnostic precision for case ascertainment (such as an ICD-10 twice coding rule[33]) or take advantage of computational linguistic techniques (eg, free-text extraction using natural languages processing approaches).[11 34]

## Implications

The work described sets a precedent for education data being used for patient or medical benefit in England. The regulatory and technical issues for data sharing between health and non-health services are challenging. Certainly, to develop and improve linked data resources, partnerships between academic and government institutions should continue to explore public opinion and develop guidance on building a 'social license' for the sustained use of linked data.[31] In addition, it is important that recent policies that support accessibility for reuse in the future are sustained, especially given the time and resources spent to set up linked data resources and the potential they hold.[35]

Record linkages are a valuable enhancement to child-based longitudinal studies and clinical registries, which allow evaluation of questions relevant to public health and social care policy. We would urge all mental health trials conducted in children that might influence their attendance or function at school to link to the NPD. We hope our experience may provide a useful guide for other health services wishing to build information resources using linked administrative data and specifically to encourage other mental health service providers to work together to link their data to NPD. In time, we hope these resources will generate a wider network of fine-grained data and analytical expertise, which can be used for research to inform commissioning and service provision and better meet children and young person's mental health needs within the population.

**Author affiliations**
[1]Institute of Psychiatry, Psychology and Neuroscience, Kings College London, London, UK
[2]NIHR South London and Maudsley NHS Foundation Trust Biomedical Research Centre, London, UK
[3]University of Exeter Medical School, Exeter, UK
[4]Evidence Based Practice Unit, UCL and Anna Freud Centre, London, UK
[5]UCL Institute of Education, University College London, London, UK
[6]Administrative Data Research Centre for England, UCL Great Ormond Street Institute of Child Health, London, UK

**Acknowledgements** The authors are very grateful to Richard White, Karen Stevens and Martin Johnson, and members of DfE National Pupil Database team, who provided invaluable support throughout the project.

**Contributors** The study was conceived by JMD, TF and MH. Data extraction was carried out by JMD with support from HS, MB, SE, RL and AJ. Data analysis was undertaken by JMD. Reporting of findings was led by JMD with support from RG, TM, SE, TF, JD and RH, supervised by RS and MH. All authors contributed to manuscript preparation and approved the final version.

**Funding** This work was supported by the Clinical Records Interactive Search (CRIS) system funded and developed by the National Institute for Health Research (NIHR)

Mental Health Biomedical Research Centre at South London and Maudsley NHS Foundation Trust and King's College London and a joint infrastructure grant from Guy's and St Thomas' Charity and the Maudsley Charity (grant number BRC-2011-10035). JD received support by a Medical Research Council (MRC) Clinical Research Training Fellowship (MR/L017105/1) and Psychiatry Research Trust Peggy Pollak Research Fellowship in Developmental Psychiatry. RDH was funded by an MRC Population Health Scientist Fellowship (grant number MR/J01219X/1). MH, RS, AJ, MB, RL and HS received salary support from the NIHR Mental Health Biomedical Research Centre at South London and Maudsley NHS Foundation Trust and King's College London. JD was supported by the NIHR Collaboration for Leadership in Applied Health Research and Care North Thames at Bart's Health NHS Trust (NIHR CLAHRC North Thames). RG and JD are members of the Policy Research Unit in the Health of Children, Young People and Families (CPRU), which is funded by the England Department of Health Policy Research Programme.

**Disclaimer** The views expressed are those of the author(s) and not necessarily those of the NHS, the NIHR or the Department of Health and Social Care.

**Competing interests** None declared.

**Patient consent for publication** Not required.

**Ethics approval** The CRIS data resource received ethical approval as an anonymised data set for secondary analyses from Oxfordshire REC C (Ref: 08/H0606/71+5) and NHS Health Research Authority Confidentiality Advisory Group, reference: CAG 9-08(a)/2014.

**Provenance and peer review** Not commissioned; externally peer reviewed.

**Data sharing statement** The data accessed by CRIS remain within an NHS firewall and governance is provided by a patient-led oversight committee. Subject to these conditions, data access is encouraged and those interested should contact RS (robert.stewart@kcl.ac.uk), CRIS academic lead.

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
