## [Reviewer comments · BMJ Open]

ARTICLE DETAILS

TITLE (PROVISIONAL)	An approach to linking education, social care and electronic health records for children and young people in South London: a linkage study of child and adolescent mental health service data.
AUTHORS	Downs, Johnny; Ford, Tamsin; Stewart, Robert; Epstein, Sophie; Shetty, Hitesh; Little, Ryan; Jewell, Amelia; Broadbent, Matthew; Deighton, Jessica; Mostafa, Tarek; Gilbert, Ruth; Hotopf, Matthew; Hayes, Richard

VERSION 1 – REVIEW

REVIEWER	Dr. S. (Stefan) Koudstaal University Medical Centre Utrecht, the Netherlands
REVIEW RETURNED	14-Jul-2018

GENERAL COMMENTS	Downs and co-workers described the forthcoming of a dataset that consists of linked electronic health records covering education, social care, and mental health services. The endeavour spanning several years of work is laudable and the manuscript describes that journey in an unsurpassed level of detail and clarity. Overall, the manuscript is well written and relevant to the emerging field of linked routine care data as a research platform. Major comment: -Odds ratio for positive linkage (Table 1) show interesting and robust trends for age, ethnicity, and ICD-10 codes reflecting mental disorders. However, why is social deprivation inconsistent, especially the least deprived group? Are the negative odds ratios in the 2nd and 3rd group explained by increasing percentages of children attending non-state maintained schools? Why is the least deprived group then equal in terms of positive linkage compared to the most deprived reference group. Thoughts on that would be a welcome addition to the Discussion section of the manuscript. Minor comments none
--

REVIEWER	Danny T.Y. Wu, PhD, MSI University of Cincinnati Department of Biomedical Informatics
REVIEW RETURNED	30-Aug-2018

GENERAL COMMENTS	This monumental paper includes novel aspects of data linkage and governance with applications in the field of mental health research. My major comments are: 1. It would be nice to have more discussion of the multiaxial system (MAS) of ICD-10 codes, or at least a citation of the material used to derive such a system; explaining its efficacy (e.g. DOI:
---

	10.1159/000285017). Additionally, it would be interesting to see how each of these axes act separately when compared to absence rates. Although the overall likelihood was 10% greater, would it be significantly different for each individual axis? 2. This article seems very much to parallel the idea of PheWAS, or phenome-wide association studies (e.g DOI: 10.1111/imm.12195) but did not mention any of them. Instead of associating genotypes and phenotypes (ICD codes), the authors associate behavioral/environmental phenotypes (absence rate) with clinical phenotypes (ICD codes). Generally, multi-effect models such as logistic regression are common in this territory as well, providing a lot of statistical literature to draw on. This could even open up a new kind of “WAS”... It would certainly be worth exploring in future work at least. 3. With #2 being said, in PheWAS data cleansing mechanisms are important. Not all postcodes or school systems may have equal access to mental health facilities; an erroneous code may be entered; etc. In PheWAS, the “rule of three” is utilized to include individuals with at least three instances of a given ICD code, while having zero ICD code instances as a control (an ICD code appearing once or twice is usually excluded in large samples). It would be important to discuss the limitations of ICD codes in these respects. 4. It would make sense to try different models of school absence binarization. The <80% school attendance makes sense, but different cutoffs (e.g. 60%, 70%, 90%) may affect the model and change the findings. 5. While confidence intervals have been included in the table, it would be better to include them in the body of the paper itself where relevant. Additionally, there are some minor suggestions:  1. (P4L56) NHS (National Health Service) is never spelled out when it's used in the first time. 2. (P5L46) this sentence is a little confusing. “...four south boroughs - as well as some specialist services...” change “as well as” to “because”? rewrite the sentence? 3. (P12L26) “both DfE and SLaM data controllers both expressed...” “, remove the second “both” 4. (P12L40) The numbers seem not matching. The sum of the percentage of the four stages (60.4% + 4.4% + 1.1% + 20.7%) is 86.6%, not 82.5%. 5. (P13L41) “Using deterministic matching techniques provided by DfE...” The word “deterministic” only appears twice in the discussion section. Should “deterministic matching” be explained in the method section and linked to the practice? 6. (P14L50) It is unclear what “flexibility” the government provided using the data separation principle in this sentence. Should this be described in the method section? 7. (P16L20) “Because the source data was available data to examine...” , remove the second “data”
--	---

VERSION 1 – AUTHOR RESPONSE

Reviewer 1, Comment 1: “Odds ratio for positive linkage (Table 1) show interesting and robust trends for age, ethnicity, and ICD-10 codes reflecting mental disorders. However, why is social deprivation inconsistent, especially the least deprived group? Are the negative odds ratios in the 2nd and 3rd group explained by increasing percentages of children attending non-state maintained schools? Why is the least deprived group then equal in terms of positive linkage compared to the most deprived

reference group. Thoughts on that would be a welcome addition to the Discussion section of the manuscript.”

Author response: this is a great point raised by the reviewer, and we have included the following as a 3rd paragraph in the Discussion.

We found a U-shaped distribution in neighbourhood deprivation and likelihood of linkage. Compared to areas with the highest deprivation, areas within the 2nd and 3rd quartiles showed significantly reduced likelihood of linkage, but the most affluent areas showed minimal difference. This could relate to families from affluent areas being able to comply with the administrative process, and/or correct administrative errors, and families from the highest deprived areas having greater need and hence higher clinical contact with services. Both these factors may improve clerical accuracy and concordance with school data. Families from 2nd and 3rd quartiles may have less of both these characteristics, and hence reduce their likelihood of linkage. The current data available in this study does not permit this hypothesis to be tested, but findings suggest that a more detailed extraction examining frequency of clinical contact with services and data linkage outcome is an area for future work.

Reviewer 2, Comment 1: “It would be nice to have more discussion of the multi-axial system (MAS) of ICD-10 codes, or at least a citation of the material used to derive such a system; explaining its efficacy (e.g. DOI: 10.1159/000285017).”

Author response: We thank the reviewer for pointing this out, and we have now included the standard reference for describing the ICD-10 multi-axial system in child and adolescent psychiatry - Rutter, Michael &

World Health Organization. (1996). Multi-axial classification of child and adolescent psychiatric disorders: the ICD-10 classification of mental and behavioural disorders in children and adolescents. Cambridge : Cambridge University Press

Reviewer 2, Comment 2: “This article seems very much to parallel the idea of PheWAS, or phenome-wide association studies (e.g DOI: 10.1111/imm.12195) but did not mention any of them. Instead of associating genotypes and phenotypes (ICD codes), the authors associate behavioral/environmental phenotypes (absence rate) with clinical phenotypes (ICD codes). Generally, multi-effect models such as logistic regression are common in this territory as well, providing a lot of statistical literature to draw on. This could even open up a new kind of “WAS”... It would certainly be worth exploring in future work at least. “

Author response: We are grateful to the reviewer for highlighting the similarities between this linkage endeavour and the PheWAS resources that are being developed internationally. However, in this submission we have been wary about drawing comparisons between the development of PheWAS approaches and our own linked data resource. The purpose of this linkage project was not to explore genetic attributes and health-education phenotypes, and our ethical and governance requirements do not permit our work to extend into this sensitive area. Hence we are cautious about any making references to genetic work and readers inferring this will be a next step in our research.

Reviewer 2, Comment 3 : In PheWAS, the “rule of three” is utilized to include individuals with at least three instances of a given ICD code, while having zero ICD code instances as a control (an ICD code appearing once or twice is usually excluded in large samples). It would be important to discuss the limitations of ICD codes in these respects.”

Author response: We are grateful to the reviewer for highlighting this methodological issue, and in response we have included the following statement as a study limitation within 9th paragraph within the discussion section of the manuscript.

ICD-10 codes permitted us to evaluate the effect of reaching threshold for a “clinical disorder” on absence rates in an efficient and cost-effective manner. However, collapsing ICD-10 categories into one binary variable only provided an ‘average’ effect across all ICD-10 diagnoses. This may have introduced aggregation bias, which disguised the potential heterogeneity of effects across different the diagnoses. Furthermore, the validity of ICD-10 codes in psychiatric registers can be variable, and although we did not disaggregate ICD-10 cases into specific disorders, it known some disorder codes are more likely to be misclassified than others, or at least more prone to diagnostic revision.¹ Assessing the effect of variation in ICD-10 validity on school outcomes was beyond the scope of this study. However, we have provided solid ground-work for future research to refine the characterisation of clinical phenotypes either via algorithms that offer greater diagnostic precision for case-ascertainment (such as an ICD-10 twice coding rule²) or take advantage of computational linguistic techniques (e.g. free-text extraction using natural languages processing approaches).^{3,4}

Reviewer 2, Comment 4. “Additionally, it would be interesting to see how each of these axes act separately when compared to absence rates. Although the overall likelihood was 10% greater, would it be significantly

different for each individual axis? It would make sense to try different models of school absence binarization. The <80% school attendance makes sense, but different cutoffs (e.g. 60%, 70%, 90%) may affect the model and change the findings.”

Author response: To address this helpful comment we also have included the following statement in the limitation section of the manuscript.

The matching evaluation also has several limitations. We only reported on a single categorical absence outcome (less than 80% annual school attendance); whether linkage error had similarly limited effects on other discrete levels of absence (e.g. 60% or 90%) was not evaluated.

Reviewer 2, Comment 5. “While confidence intervals have been included in the table, it would be better to include them in the body of the paper itself where relevant.”

Author response: Yes, C.I. and p values have been added in relevant areas within the results section.

Reviewer 2 minor suggestions:

1. (P4L56) NHS (National Health Service) is never spelled out when it's used in the first time.

Author response: amended as suggested

2. (P5L46) this sentence is a little confusing. "...four south boroughs - as well as some specialist services..." change "as well as" to "because"? rewrite the sentence?

Author response: amended to demonstrate more clearly how SLaM is the exclusive provider to the local area, and also takes referrals from other areas of the countries for more specialist services.

3. (P12L26) "both DfE and SLaM data controllers both expressed..." , remove the second "both"

Author response: amended as suggested

4. (P12L40) The numbers seem not matching. The sum of the percentage of the four stages (60.4% + 4.4% + 1.1% + 20.7%) is 86.6%, not 82.5%.

Author response: amended to reflect the matching rates of 82.5% with school absence dataset.

5. (P13L41) "Using deterministic matching techniques provided by DfE..." The word "deterministic" only appears twice in the discussion section. Should "deterministic matching" be explained in the method section and linked to the practice?

Author response: To address this, we have added a short explanatory section in paragraph 2 of the introduction.

Deterministic linkage describes an approach when a set of predetermined rules are used to classify pairs of records as matched or nonmatched. These tend to require an exact or partial agreement on a set of personal identifiers for example a successful match on the first name or surname, and match on both the date of birth and postcode. Strict deterministic methods are straightforward to use and commonly employed in government departments, however they can create high levels of missed matches between records.5As a consequence, this undermines the confidence that all the relevant records for an individual have been accurately combined across the different data sources.

6. (P14L50) It is unclear what "flexibility" the government provided using the data separation principle in this sentence. Should this be described in the method section?

Author response: We referred to flexibility in the 'governance' arrangements, however we appreciate this could be better clarified so we have added a short phrase in paragraph 4 of the discussion, to help.

In our case, by permitting approved NHS researches to examine the identifier fields of matched and unmatched SLaM samples, this governance has enabled some flexibility with the 'data separation principle' : a common practice in data linkage research, where identifiers (e.g. names or date of birth) are kept separate from attributes (in this case health or education data), to protect privacy and avoid disclosure during the linkage process.30

7. (P16L20) "Because the source data was available data to examine..." , remove the second "data"

Author response: amended as suggested

Your sincerely

Dr Johnny Downs
Peggy Pollak Fellow in Neurodevelopmental Psychiatry, KCL
Honorary Consultant in Child and Adolescent Psychiatry

References:

- 1 Davis KAS, Sudlow CLM, Hotopf M. Can mental health diagnoses in administrative data be used for research? A systematic review of the accuracy of routinely collected diagnoses. *BMC Psychiatry* 2016; 16: 263.
- 2 Hebring SJ. The challenges, advantages and future of phenome-wide association studies. *Immunology* 2014; 141: 157–65.
- 3 Downs J, Dean H, Lechler S, et al. Negative Symptoms in Early-Onset Psychosis and Their Association With Antipsychotic Treatment Failure. *Schizophr Bull* 2018. DOI:10.1093/schbul/sbx197.
- 4 Perera G, Broadbent M, Callard F, et al. Cohort profile of the South London and Maudsley NHS Foundation Trust Biomedical Research Centre (SLaM BRC) Case Register: current status and recent enhancement of an Electronic Mental Health Record-derived data resource. *BMJ Open* 2016; 6: e008721.
- 5 Harron K, Goldstein H, Dibben C. *Methodological Developments in Data Linkage*. Chichester, West Sussex, United Kingdom: Wiley-Blackwell, 2015.

VERSION 2 – REVIEW

REVIEWER	Dr S. Koudstaal UMCU Utrecht the Netherlands
REVIEW RETURNED	11-Oct-2018
GENERAL COMMENTS	I have no further comments.